# Carvacrol Inhibits Expression of Transient Receptor Potential Melastatin 7 Channels and Alleviates Zinc Neurotoxicity Induced by Traumatic Brain Injury

**DOI:** 10.3390/ijms232213840

**Published:** 2022-11-10

**Authors:** Minwoo Lee, Song Hee Lee, Seunghyuk Choi, Bo Young Choi, Sang Won Suh

**Affiliations:** 1Department of Physiology, Hallym University, College of Medicine, Chuncheon 24252, Korea; 2Department of Neurology, Hallym Neurological Institute, Hallym University Sacred Heart Hospital, Anyang 14068, Korea; 3Department of Physical Education, Hallym University, Chuncheon 24252, Korea; 4Institute of Sport Science, Hallym University, Chuncheon 24252, Korea

**Keywords:** traumatic brain injury, zinc, neuronal death, transient receptor potential melastatin 7, carvacrol

## Abstract

Carvacrol is a monoterpenoid phenol produced by aromatic plants such as oregano. Although the exact mechanism by which carvacrol acts has not yet been established, it appears to inhibit transient receptor potential melastatin 7 (TRPM7), which modulates the homeostasis of metal ions such as zinc and calcium. Several studies have demonstrated that carvacrol has protective effects against zinc neurotoxicity after ischemia and epilepsy. However, to date, no studies have investigated the effect of carvacrol on traumatic brain injury (TBI)-induced zinc neurotoxicity. In the present study, we investigated the therapeutic potential of carvacrol for the prevention of zinc-induced neuronal death after TBI. Rats were subjected to a controlled cortical impact, and carvacrol was injected at a dose of 50 mg/kg. Histological analysis was performed at 12 h, 24 h, and 7 days after TBI. We found that carvacrol reduced TBI-induced TRPM7 over-expression and free zinc accumulation. As a result, subsequent oxidative stress, dendritic damage, and neuronal degeneration were decreased. Moreover, carvacrol not only reduced microglial activation and delayed neuronal death but also improved neurological outcomes after TBI. Taken together, these findings suggest that carvacrol administration may have therapeutic potential after TBI by preventing neuronal death through the inhibition of TRPM7 expression and alleviation of zinc neurotoxicity.

## 1. Introduction

Traumatic brain injury (TBI) is a major cause of death and disability in patients with trauma [1]. It accounts for almost 50 million cases and more than 4.7 million mortalities annually [2]. TBI contributes to neuronal loss and subsequent cognitive dysfunction, even in mild to moderate injuries. Numerous studies have focused on targeting various secondary injury cascades, in order to alleviate TBI-induced cell death and tissue degeneration. Nevertheless, most studies have demonstrated limited translational success, and trials testing the effect of multiple medications have so far been unsuccessful [3].

TBI-induced neuronal death results from a cascade of events involving the derangement of glutamate and zinc homeostasis. Although zinc is essential for the functioning of numerous enzymes and DNA-binding transcription factors, excessive zinc accumulation ultimately leads to neuronal damage. Most zinc ions are tightly bound by proteins, whereas chelatable zinc ions are present in the axonal terminals of glutamatergic neurons in the hippocampus and cerebral cortex [4,5]. Chelatable zinc lies in the synaptic vesicles of pre-synaptic terminals and is released upon depolarization into the extracellular space [6]. Free zinc released from the pre-synaptic terminal acts as a neuromodulator during synaptic transmission [7]. On the other hand, under pathologic conditions such as cerebral ischemia, hypoglycemia, epilepsy, and TBI, the excessive translocation of synaptic zinc can induce neuronal damage in post-synaptic neurons [8].

Transient receptor potential melastatin 7 (TRPM7), a non-selective cation channel [9], is generally permeable to metal ions, exhibiting particularly high permeation of Zn^2+^ but also permeating other essential divalent cations such as Mg^2+^ and Ca^2+^ [10,11]. Under various pathological conditions, extracellular Ca^2+^ and Zn^2+^ flow into the intracellular space through TRPM7 and produce reactive oxygen species (ROS). The ROS then activate both membrane-bound TRPM7 and intracellular-vesicle-bound TRPM7 [12], resulting in the excess accumulation of free zinc in the cytoplasm, which leads to devastating neuronal injury [13]. Thus, the inhibition of TRPM7-mediated zinc translocation could be a potential target for TBI.

Carvacrol is a monoterpenoid phenol produced by aromatic plants such as oregano. A previous study has shown that carvacrol inhibited the function of TRPM7 ectopically expressed in a primary culture of hippocampal cornu ammonis (CA)1-3 neurons [14]. Several studies have shown that carvacrol has neuroprotective effects in several neurological disorders, including TBI [15], epilepsy [16], and hemi-Parkinsonism [17]. However, most previous studies have hypothesized that the neuroprotective effects of carvacrol are due to the decrement of calcium influx, while the role of carvacrol in TBI-induced zinc neurotoxicity has not yet been well elucidated.

In this regard, we hypothesize that carvacrol inhibits TBI-induced zinc translocation through the prevention of TRPM7 over-expression, resulting in a reduction in subsequent neuronal death and microglial activation. The carvacrol-mediated inhibition of TBI-induced TRPM7 over-expression was observed to be associated with the reduction in neuronal death and microglial activation, in a manner dependent on TRPM7-mediated intracellular free zinc accumulation. Furthermore, we found that carvacrol improved TBI-induced neurological deficits. Taken together, these findings suggest that carvacrol treatment could reduce neuronal death, subsequently ameliorating behavioral deficits after TBI.

## 2. Results

### 2.1. Carvacrol Treatment Attenuated Over-Expression of TRPM7 and Accumulation of Free Zinc in the Hippocampal Neurons after TBI

To evaluate the effect of carvacrol on TRPM7 expression after TBI, rats were given an intraperitoneal injection of carvacrol (50 mg/kg) immediately after TBI. At 12 h after TBI, we performed co-staining of neuronal nuclei (NeuN) and TRPM7 antibodies, in order to analyze the expression of TRPM7 in hippocampal neurons. Immunohistochemical examination of the hippocampal CA1 and dentate gyrus (DG) areas showed the expression of NeuN or TRPM7 in the sham and TBI groups (Figure 1A). We found that the TRPM7 fluorescence intensity was greater in the TBI group than in the sham-operated groups. However, the TRPM7 fluorescence intensity was significantly decreased in the carvacrol-treated TBI group, compared to the TBI group that received the vehicle (Figure 1B–D). Western blot analysis also indicated the up-regulation of TRPM7 proteins in the ipsilateral hippocampus after TBI, compared to the sham-operated groups. However, the protein amount of TRPM7 was significantly reduced in the carvacrol-treated TBI group, compared to the vehicle-treated TBI group (Figure 1E,F).

As other studies have reported, TBI induces the excessive release of zinc from synaptic boutons. The released zinc can enter the intracellular space through TRPM7 [18], thereby causing the accumulation of zinc in post-synaptic neurons. In addition, TRPM7 is also known to be abundant in intracellular vesicles (M7Vs) that contain zinc ions, leading to the translocation of zinc from the vesicle to the cytoplasm [12]. As such, we wondered whether carvacrol affects the TRPM7-mediated zinc accumulation after TBI. To determine the impact of carvacrol on the accumulation of intracellular free zinc after TBI, we conducted TSQ (6-methoxy-8-p-toluene sulfonamide-quinoline) staining; in particular, TSQ is a specific indicator of intracellular free zinc levels. Compared to the vehicle-treated groups, the number of TSQ fluorescence neurons was significantly reduced in the carvacrol-treated groups after TBI (Figure 1E–G). Taken together, these data demonstrate that carvacrol treatment reduced the TRPM7 over-expression and intracellular free zinc accumulation after TBI.

### 2.2. Carvacrol Treatment Reduced TBI-Induced Neuronal Injury

As carvacrol administration resulted in the inhibition of TRPM7 over-expression and zinc accumulation, we assessed the effect of carvacrol on the prevention of TBI-induced neuronal injury. First, we performed Fluro-Jade B (FJB) staining in brains harvested 24 h after TBI to detect degenerating neurons. TBI administration led to remarkable neuronal degeneration. FJB-stained cells emerged in the CA1 and DG regions of the ipsilateral hippocampus (Figure 2A,C). The number of FJB^+^ cells in the carvacrol-treated group was significantly lower than in the vehicle-treated group (Figure 2B,D).

Next, we assessed dendritic damage in the ipsilateral hippocampus using Microtubule-associated protein 2 (MAP-2) immunofluorescence staining. As previously reported, TBI causes a significant loss of dendritic proteins—a characteristic of dendritic damage—within 24 h after the insult [19]. Sham-operated groups showed neutral MAP-2^+^ expression in dendrites of the hippocampal CA1, GCL, and hilus. Mice between vehicle- and carvacrol-treated groups showed no differences in fluorescence intensity and %area of MAP-2 immunoreactivity (IR). Both intensity and %area of MAP-2 IR were remarkably decreased 24 h after TBI, compared with the sham-operated groups. However, we observed a significant increase in MAP-2 IR in the carvacrol-treated TBI group, compared with the vehicle-treated TBI group. A similar trend was seen for the MAP-2 %area (Figure 3). Taken together, these data suggest that carvacrol treatment reduced neuronal damage after TBI.

### 2.3. Carvacrol Treatment Reduced TBI-Induced Oxidative Stress

TBI is well-known to induce oxidative stress by increasing the generation of reactive oxygen and nitrogen species (ROS/RNS) and subsequent lipid peroxidation [20]. We hypothesized that carvacrol could reduce TBI-induced oxidative stress. First, we assessed lipid peroxidation in the ipsilateral hippocampus using immunofluorescence staining. The molecule 4-HNE, the major end-product of lipid peroxidation, has been widely accepted as an inducer of oxidative stress [21,22]. TBI led to an increase in 4-HNE fluorescence in the hippocampal CA1, GCL, and hilus, while 4-HNE fluorescence intensity was significantly attenuated by carvacrol treatment (Figure 4A–D). Next, we examined the effect of carvacrol on nitrotyrosine formation and NO production caused by TBI. We observed the fluorescence signal of nNOS and nitrotyrosine in the hippocampal CA1 after TBI, which was significantly reduced by carvacrol treatment (Figure 4E–G). Furthermore, as oxidative stress can also significantly alter the cellular redox status by depleting cellular sulfhydryl compounds, we evaluated whether carvacrol affected neuronal glutathione (GSH). Sections were histologically stained by probing for GSH-N-ethylmaleimide (NEM) adducts (GS-NEM), in order to detect the reduced form of GSH in neurons. There were no significant differences in GS-NEM IR between vehicle- and carvacrol-treated sham groups. Compared with the sham-operated groups, GS-NEM IR in the ipsilateral CA1 was significantly increased in the vehicle-treated group 24 h after TBI. In contrast, the carvacrol-treated TBI group revealed highly reduced IR to GS-NEM, compared to the vehicle-treated TBI group (Figure 4H,I). Taken together, these data imply that carvacrol treatment decreased oxidative stress after TBI.

### 2.4. Carvacrol Treatment Reduced TBI-induced Microglial Activation

TBI induces several pathophysiological changes, such as inflammatory and immune responses by activated microglia. As TRPM7 is also expressed in microglia, we examined the effect of carvacrol on microglial activation caused by TBI. To confirm this effect of carvacrol, we first performed CD11b immunostaining. The CD11b signal consistently showed that the vehicle-treated TBI group, but not the sham-operated groups or carvacrol-treated TBI group, presented obvious microglial activation in the ipsilateral hippocampus (Figure 5A,B). Next, we assessed whether carvacrol could affect their polarization states after TBI. Ionized calcium-binding adaptor molecule 1 (Iba-1) is constitutively expressed in microglia. Staining against the microglia marker Iba-1 revealed similar patterns to those observed in CD11b staining. We further assessed the co-localization of CD68, a lysosome-associated membrane protein, and Iba-1, which is well-known to be an indicator of M1-polarized microglia [23] and to functionally regulate phagocytosis [24]. We found that most of the Iba-1-positive microglia remarkably expressed CD68 at 7 days after TBI (Figure 5C). In contrast, carvacrol treatment not only reduced Iba-1 (Figure 5D) and CD68 (Figure 5E) immunoreactivity, but also significantly decreased their co-localization (Figure 5F,G) in TBI groups. Taken together, these data indicate that carvacrol treatment reduced the M1 activation states of microglia through TRPM7 inhibition.

### 2.5. Carvacrol Treatment Improved Neurological Impairment and Reduced Neuronal Death after TBI

To determine whether carvacrol treatment could alleviate neurological dysfunction after TBI, we assessed the changes in the modified neurological severity score (mNSS) and calculated the variation in ΔmNSS over time. Following TBI, neurological deficits were evaluated at 1, 24, 48, and 72 h, as well as on the seventh day. Carvacrol treatment resulted in improved neurological function, as demonstrated by a decrease in mNSS (Figure 6A and Table 1) and an increase in ΔmNSS scores (Figure 6B), compared to the vehicle-treated group. As carvacrol improved neurological function, we also examined whether carvacrol could prevent TBI-induced neuronal death. Sham-operated groups showed intense staining of NeuN^+^ neurons in CA1 and DG. The number of NeuN^+^ stained neurons was decreased 7 days after TBI, compared to the sham-operated groups; however, the number of NeuN^+^ neurons in the carvacrol-treated group was significantly higher than that in the vehicle-treated group (Figure 6C–E). Taken together, these findings show that carvacrol treatment significantly prevented TBI-induced neuronal death and neurological impairment.

### 2.6. 2-APB or NS8593 Treatment Reduced TRPM7 Over-expression, Free Zinc Accumulation, and Neuronal Injury after TBI

To assess whether treatment with 2-APB or NS8593—another TRPM7 inhibitor—could reduce the over-expression of TRPM7, accumulation of intracellular free zinc, and degeneration of neurons after TBI, 2-APB or NS8593 was injected intraperitoneally at a dose of 2 mg/kg or 5 mg/kg immediately following TBI. The 2-APB and NS8593 treatments significantly attenuated TBI-induced TRPM7 over-expression, compared to the vehicle-treated TBI group (Figure 7A–C). In addition, the number of TSQ fluorescence neurons (Figure 7D–F) and degenerating neurons (Figure 7G–J) was remarkably reduced in the 2-APB- or NS8593-treated groups after TBI, compared to the vehicle-treated group. These findings indicate that 2-APB or NS8593 treatment also reduces the accumulation of intracellular free zinc by inhibiting the over-expression of TRPM7, thereby leading to a significant reduction in neuronal degeneration.

## 3. Discussion

In the present study, we observed that carvacrol administration resulted in a reduction in delayed neuronal loss and better neurological outcomes after TBI, mediated by decreased zinc accumulation through the inhibition of TRPM7. The initial pathological mechanisms of TBI are characterized by direct neuronal and glial damage, BBB disruption, and subsequent cerebral ischemia [25]. These primary injuries result in secondary neuronal injury hours to months later, by means of multiple cascades such as modified zinc and calcium homeostasis, neuroinflammation, and oxidative stress. These pathological alterations can manifest as delayed motor and balance deficits, cognitive impairment, and other neurological symptoms in both human and experimental animals [26,27]. We hypothesized that carvacrol could alleviate secondary neuronal injury after TBI, through the prevention of devastating cascades in neurons and glial cells.

Our previous studies considering multiple neurological disease models (e.g., TBI, global cerebral ischemia, epilepsy, and hypoglycemia) have revealed that such insults invariably induce excessive zinc translocation into hippocampal neurons [28,29,30,31]. After TBI, excessive free zinc ions translocate into post-synaptic neurons through cation channels and multiple transporters including TRP family, AMPA/kainite glutamate receptors, and NMDA receptors [18,32,33,34]. Of these, TRPM7 is a class of TRP channel that regulates Zn^2+^, Ca^2+^, and Mg^2+^ homeostasis, as well as the release of neurotransmitters [18]. TRPM7 is essential for normal organ development, but may also trigger anoxic neuron death [13]. TRPM7 channels are present both in post-synaptic membranes and intracellular vesicles called M7Vs, which accumulate Zn^2+^ in a glutathione-enriched lumen when cytosolic Zn^2+^ concentrations are elevated [12]. Increased production of ROS after TBI is sensed by both TRPM7 channels, leading to the synergistic accumulation of free zinc ions in the cytoplasm, resulting in positive feedback for the production of further ROS, which aggravates neuronal death. A potential pathophysiological link between ROS and Zn^2+^ release by TRPM7 involves the amount of cytosolic GSH. A previous study has shown that GSH, not glutathione disulfide (GSSG), prevents Zn^2+^ influx through TRPM7 in a dose-dependent manner [12]. As initial traumatic injury causes oxidative stress, the GSH/GSSG ratio decreases, which may result in further Zn^2+^ influx after TBI. Here, we found that the GS-NEM was significantly less decreased in carvacrol-treated groups, implying that the reduced form of glutathione was less-consumed through the inhibition of TRPM7 channels. Thus, we hypothesized that inhibition of the TRPM7 channel would result in a decrement in post-synaptic zinc accumulation, thus alleviating zinc-induced neuronal death.

Carvacrol, a TRPM7 inhibitor, is a monoterpenoid phenol that is present in the natural oil of oregano [14]. Our group has previously assessed the therapeutic potentials of carvacrol in acute neurological disorders, including global cerebral ischemia [35] and epilepsy models [36] (Table 2). The common denominator of these models and TBI is the excessive zinc translocation-induced secondary neuronal injuries, though the mode and pathophysiology of the initial neurological insults differ. In this regard, we aimed to assess the effects of carvacrol administration in TBI models, in order to expand the current knowledge by adding more insights into the pathophysiological cascades and functional outcomes. TRPM7 has previously been studied as a potential target for the prevention of TBI-induced neuronal damage, with a focus on calcium influx decrement [15]. The TRPM7 inhibitor has been shown to reduce TBI-induced neuronal injury through several cascades, including a decrease in calcium influx, followed by a reduction in lactate dehydrogenase release, apoptosis, and caspase-3 activation. While Zn^2+^ is known to have the highest affinity to TRPM7 channels among the various divalent cations [37], the role of TRPM7 in free zinc translocation after TBI has not yet been fully elucidated. Previous studies have revealed that the primary divalent cation for TRPM7 channels is Zn^2+^, rather than Ca^2+^, and TRPM7 regulates Zn^2+^ levels in cultured hippocampal neurons [11]. Furthermore, previous studies have revealed that excessive Zn^2+^ accumulation after TBI is neurotoxic [38]. While carvacrol is known to inhibit TRPM7 channels, our findings revealed that carvacrol reduced the TRPM7 protein expression, which was increased after TBI. The mechanism by which carvacrol inhibits TRPM7 protein expression has not yet been fully elucidated. Nonetheless, we speculate that carvacrol may not directly cause the down-regulation of TRPM7 protein expression but, instead, may suppress further rounds of up-regulation after initial TRPM7 expression by TBI. These observations and our findings suggest that utilizing carvacrol to inhibit TRPM7 channels and the subsequent decrement of zinc influx into the intracellular space may serve as a potential therapeutic strategy to reduce TBI-induced neuronal death (Figure 8A,B). 

Excessive glial activation due to TBI also contributes to further neuronal injuries. Microglia are the major glial cells in the brain that help to maintain the homeostasis of the central nervous system through the secretion of neurotrophic factors [29,30]. However, the excessive and rapid activation of microglia leads to secondary neuronal damage through pro-inflammatory and immune responses after neurological insults, including TBI [39]. Activated microglia initiate morphological changes into an amoeboid pattern, migrating to the injured site and releasing a number of neurotoxic substances such as RNS/ROS, pro-inflammatory cytokines, and metalloproteinases (MMPs) [40,41]. In the setting of TBI, microglial activation can be triggered by excessive extracellular zinc influx into resting-state microglia through the microglial TRPM7 channels (Figure 8C,D). Our study revealed that most microglial activation was confined to M1-polarized microglia, which are mediators of proinflammatory responses [42]; namely, carvacrol significantly decreased M1 microglial activation.

There are limitations to be addressed. First, none of the TRPM7 inhibitors utilized in this study—including carvacrol, 2-APB, and NS8593—are highly specific inhibitors of the TRPM7 channel. We could not prove the potential impact of other downstream activities with carvacrol. However, specific TRPM7 inhibitors are currently not available. Further, as deletion of TRPM7 in whole animals results in death, we could not conduct experiments using knockout mice. To confirm and further strengthen our hypothesis, the development of a specific TRPM7 inhibitor is needed.

In summary, the present study demonstrated that carvacrol treatment reduced the over-expression of TRPM7 and the accumulation of free zinc in hippocampal neurons after TBI. Consequently, the decrement of free zinc accumulation led to a reduction in the number of degenerating neurons, dendritic damage, oxidative stress, and glutathione depletion after TBI. Furthermore, we found that carvacrol treatment not only reduced microglial activation and delayed neuronal death, but also improved neurological outcomes after TBI. Therefore, the attenuation of TBI-induced neuronal death by carvacrol suggests that the inhibition of TRPM7-mediated intracellular free zinc accumulation has the potential to result in therapeutic benefits for TBI treatment.

## 4. Materials and Methods

### 4.1. Ethical Statement and Experimental Animals

Animal use and relevant experimental procedures were approved by the Institutional Animal Care and Use Committee of Hallym University (Protocol # Hallym 2019-68). This manuscript was written up according to the Animal Research: Reporting In Vivo Experiments guidelines. In the present study, we used male Sprague–Dawley rats (2–3 months old, 280–320 g; Daehan Biolink, Chungcheongbuk, Korea). The animals were housed under a maintained temperature (22 ± 2 °C) and humidity (55 ± 5%), in an automatically controlled 12 h light and dark cycle (lights turned on and off), with proper food (Purina, Gyeonggi, Korea) and water available ad libitum.

### 4.2. Controlled Cortical Impact for TBI

A controlled cortical impact (CCI) model of TBI was performed. In shorts, rats were anesthetized with 3% inhaled isoflurane in a 30:70 mixture of oxygen and nitrous oxide using an isoflurane vaporizer (VetEquip, Livermore, CA, USA), positioned in a stereotaxic apparatus (David Kopf Instruments, Tujunga, CA, USA), and maintained on 1.5% isoflurane. A craniotomy was performed approximately 5 mm over the right hemisphere using a drill (2.8 mm lateral to the midline and 3 mm to the bregma). With a CCI device (Leica Impact One; Leica Biosystems, Nussloch, Germany), a 3 mm flat-tip impactor was accelerated down to a 3 mm depth at a velocity of 5 m/s. All rats were kept at a core temperature of 36–37.5 °C with a homeothermic blanket control unit (Harvard Bioscience, Holliston, MA, USA) during and after surgery, until ambulatory. Injuries were evaluated at 12 h, 24 h, and 7 days post-injury. Animals were randomly assigned to TBI according to an online randomization tool (randomizer.org). Sham-operated groups only underwent craniotomy.

### 4.3. Drug Treatment and Experimental Design

We intraperitoneally administered carvacrol (catalog no. 282197; Sigma-Aldrich Co., St. Louis, MO, USA), dissolved in 0.1% DMSO, to mice at a dose of 50 mg/kg/day for the entire experimental course, starting immediately after brain injury. Control rats were injected intraperitoneally with equal volumes of 0.1% DMSO (diluted with saline) only (vehicle). To assess whether carvacrol reduced TBI-induced neuronal death, the experimental groups were divided into the following periods: in phase 1, carvacrol was injected once, and the rats were sacrificed 12 h after TBI. To analyze the expression of TRPM7 and intracellular accumulation of free zinc, the rats were divided into four groups: (1) a vehicle-treated sham group (*n* = 3), (2) carvacrol-treated sham group (*n* = 3), (3) vehicle-treated TBI group (*n* = 9), and (4) carvacrol-treated TBI group (*n* = 8). In phase 2, carvacrol was injected once, and the rats were sacrificed 24 h after TBI. To determine oxidative stress and neuronal injury, the rats were divided into four groups: (1) a vehicle-treated sham group (*n* = 3), (2) carvacrol-treated sham group (*n* = 3), (3) vehicle-treated TBI group (*n* = 6), and (4) carvacrol-treated TBI group (*n* = 5). In phase 3, carvacrol was injected once per day for 7 days starting immediately post-TBI, and the rats were sacrificed 7 days after TBI. To investigate the effect of carvacrol on TBI-induced microglial activation, delayed neuronal death, and neurological function, the rats were divided into four groups: (1) a vehicle-treated sham group (*n* = 5), (2) carvacrol-treated sham group (*n* = 5), (3) vehicle-treated TBI group (*n* = 5), and (4) carvacrol-treated TBI group (*n* = 6). In addition, in order to evaluate whether 2-APB or NS8593 treatment reduced TRPM7 expression, free zinc accumulation, and neuronal degeneration after TBI, 2-APB or NS8593 was intraperitoneally injected at a dose of 2 mg/kg or 5 mg/kg immediately following TBI, and the brain was obtained 12 or 24 h after TBI. 2-APB was dissolved in 0.9% saline, and NS8593 was kept as a stock solution of 100 Mm dissolved in DMSO and diluted in saline (1 mg/mL) before injection. Control rats were administered with equal volumes of 0.9% saline only (vehicle). The 2-APB or NS8593 concentration was determined according to previous research [36,43,44].

### 4.4. Zinc Staining

To examine the effect of carvacrol on the intracellular accumulation of free zinc after TBI, fresh frozen brain slices were reacted with *N*-(6-methoxy-8-quinolyl)-para-toluenesulfonamide (TSQ; Molecular Probes, Eugene, OR, USA) [45]. The rats were anesthetized with 4–5% isoflurane, and the brain sample was obtained without perfusion. The obtained whole brain was put on dry ice for rapid freezing and stored at –80 °C in a freezer. The frozen brain was then coronally sliced at a 20 μm thickness, mounted on gelatin-coated slides (Fisher Scientific, Pittsburgh, PA, USA), and then reacted with a TSQ solution containing 4.5 μM TSQ, 140 mM sodium barbital, and 140 mM sodium acetate for 1 min. Thereafter, the sample slides were washed with 0.9% saline for 1 min, observed with a fluorescence microscope (Olympus upright microscope, epi-illuminated with 360 nm UV light), and photographed through a 500 nm long-pass filter using an INFINITY3-1 CCD cooled digital color camera (Lumenera Co., Ottawa, ON, Canada) using the INFINITY Analyze software (release version 6.0). Quantification of the number of TSQ^+^ cells was performed by a blind observer, as described previously [46].

### 4.5. Tissue Preparation

Rats were anesthetized using a single intraperitoneal injection of urethane (concentration: 1.5 g/kg; injection volume: 0.01 mL/g body weight) dissolved in saline (0.9% NaCl). The deeply anesthetized rats were transcardially perfused with saline, followed by 4% paraformaldehyde (PFA) in phosphate-buffered saline (PBS). Their brains were harvested and post-fixed in 4% PFA for 1 h. After fixation, the brains were immersed in 30% sucrose, which served as a cryoprotectant. Thereafter, the brains were sectioned serially at 30 μm using a cryostat microtome (CM1850; Leica, Wetzlar, Germany).

### 4.6. Evaluation of Hippocampal Degenerating Neurons

To evaluate the effect of carvacrol on TBI-induced neuronal degeneration, we performed Fluoro-Jade B (FJB) staining. Brain slices from all experimental and sham groups were mounted on gelatin-coated slides and then reacted with 100% ethanol, followed by 70% ethanol, and then distilled water for hydration. Then, the samples were reacted with 0.06% potassium permanganate solution for 15 min, washed with distilled water once again for 1 min, and then soaked in 0.001% FJB solution (Histo-Chem Inc., Jefferson, AR, USA) for 30 min. Finally, the samples were hydrated with distilled water for 1 min three times. After washing, the slides were dried gently using an airflow machine (Daihan Labtech, Gyeonggi, Korea) for approximately 3–4 h. The FJB-reacted slides were then dehydrated in xylene and mounted with mounting media DPX (Sigma-Aldrich Co., St. Louis, MO, USA). The stained slides were observed using a confocal microscope (LSM 710; Carl Zeiss, Oberkochen, Germany) with excitation and emission wavelengths of 480 nm and 525 nm, respectively. A blinded experimenter precisely counted the number of FJB^+^ neurons in the ipsilateral CA1 and DG areas by obtaining five coronal sections (2.92 mm to 4.56 mm from bregma to caudal). Data are represented as the average number of degenerating neurons per region.

### 4.7. Immunohistochemistry

Sections were washed 3 times in PBS for 10 min and then immersed in 1.2% hydrogen peroxide for 20 min at room temperature, in order to block endogenous peroxidase activity. After washing, the sections were incubated in mouse monoclonal anti-NeuN antibody (diluted 1:500; Millipore, Cambridge, UK) in PBS containing 0.3% Triton X-100 at 4 °C overnight, in order to assess neuronal loss after TBI. After washing in PBS, the sections were incubated with biotinylated anti-mouse IgG (diluted 1:250; Vector, Burlingame, CA, USA) to detect NeuN antibody for 2 h at room temperature. Thereafter, sections were immersed in avidin–biotin–peroxidase complex (Vector) for 2 h at room temperature. Between incubations, the sections were washed with PBS. The immune reaction was visualized with 3,3′-diaminobenzidine (Sigma-Aldrich) in 0.01 M PBS containing 0.015% H_2_O_2_, and the sections were mounted on gelatin-coated slides and cover-slipped with Canada Balsam (Junsei Chemical Co., Ltd., Chuo-ku, Tokyo, Japan). The immunoreactions were observed under an Olympus IX70 inverted microscope. To evaluate non-specific effects, a few sections were incubated in a buffer without any primary antibody. This procedure always resulted in a complete lack of immunoreactivity. To quantify the number of NeuN^+^ neurons, five coronal sections were analyzed by a blinded experimenter using the ImageJ software (National Institutes of Health, Bethesda, MD, USA). Data are expressed as the average number of NeuN^+^ cells per region.

### 4.8. Immunofluorescence Analysis

Immunofluorescence labeling was performed according to routine immunostaining protocols such as those referenced above. The primary antibodies used in this study were as follows: Rabbit anti-TRPM7 (diluted 1:400; Alomone Labs, Jerusalem, Israel), mouse anti-4HNE (diluted 1:500; Alpha Diagnostic Intl. Inc., San Antonio, TX, USA), rabbit anti-nitrotyrosine (diluted 1:500; Abcam, Cambridge, UK), goat anti-Iba1 (diluted 1:500; Abcam), mouse anti-CD68 (diluted 1:100; Bio-Rad, Hercules, CA, USA), mouse anti-glutathione: N-ethylmaleimide adduct (GS-NEM) antibody (diluted 1:100, Millipore), mouse anti-MAP2 (diluted 1:200; Millipore), mouse anti-CD11b (diluted 1:500; Bio-Rad), and mouse anti-NeuN (diluted 1:500; Millipore). For double-labeling, primary antibodies were simultaneously incubated and further processed for each antibody. For TRPM7, NeuN, MAP2, 4HNE, nitrotyrosine, GS-NEM, CD11b, Iba-1, and CD68, fluorescent-conjugated secondary antibodies were applied (diluted 1:250; Invitrogen). Sections were counterstained with DAPI (4,6-diamidino-2-phenylindole; diluted 1:1000; Invitrogen). Fluorescence-stained sections were mounted on gelatin-coated slides and cover-slipped with DPX (Sigma-Aldrich). Fluorescence signals were detected using a Zeiss LSM 710 confocal microscope (Carl Zeiss, Oberkochen, Germany) with a sequential scanning mode for DAPI and Alexa 488 and 594. Stacks of images (1024 × 1024 pixels) from consecutive slices of 0.5–0.8 μm thickness were obtained by averaging fifteen scans per slice, and were processed using the ZEN 2 software (blue edition, Carl Zeiss). Images were taken from the hippocampal region. The quantification of the mean intensity and co-localization experiments was performed using ZEN 2 (blue edition, Carl Zeiss). The overlap coefficient (Mander’s coefficient) was used as the co-localization coefficient. The area of immunoreactivity was measured using ImageJ (National Institutes of Health), and expressed as the percentage area. In addition, five coronal sections from each rat were scored by a blinded experimenter, in order to quantify microglial activation. The criteria for microglial activation were based on the number, intensity, and morphology of CD11b^+^ cells [47].

### 4.9. Modified Neurological Severity Score

To examine whether carvacrol treatment alleviated TBI-induced neurological dysfunction, neurological examination was performed using a modified neurological severity score (mNSS), as described previously [48]. These tests were performed at 1 h, 24 h, 48 h, and 72 h, and 7 days after TBI. The mNSS includes motor (muscle power, abnormal movement), sensory (visual, tactile, and proprioceptive), balance, and reflex exams, and is graded from 0 to 18 (0 = normal function, 18 = maximal deficit). A detailed description of the mNSS test has been previously reported [49]. In short, this functional test consists of (1) raising the rat by the tail and recording flexion (3 points), (2) walking on the floor (3 points), (3) sensory test (2 points), (4) beam balance tests (6 points), and (5) reflex absence/abnormal movement tests (4 points). Each point is scored for an inability to perform a particular task or for the lack of a reflex. Thus, higher scores imply worse neurological status.

### 4.10. Statistical Analysis

All data are expressed as the mean ± SEM. Comparisons between the vehicle- and carvacrol-treated groups were performed with a two-tailed unpaired Student’s *t*-test. To compare the values among the four groups, the remaining data were analyzed using a Kruskal–Wallis test with post-hoc analysis using a Bonferroni correction. In addition, behavioral data over time among groups were analyzed by a repeated measures ANOVA using the SPSS software (ver. 21). *p*-values less than 0.05 were considered to indicate statistical significance (i.e., *p* < 0.05).

## Figures and Tables

**Figure 1 ijms-23-13840-f001:**
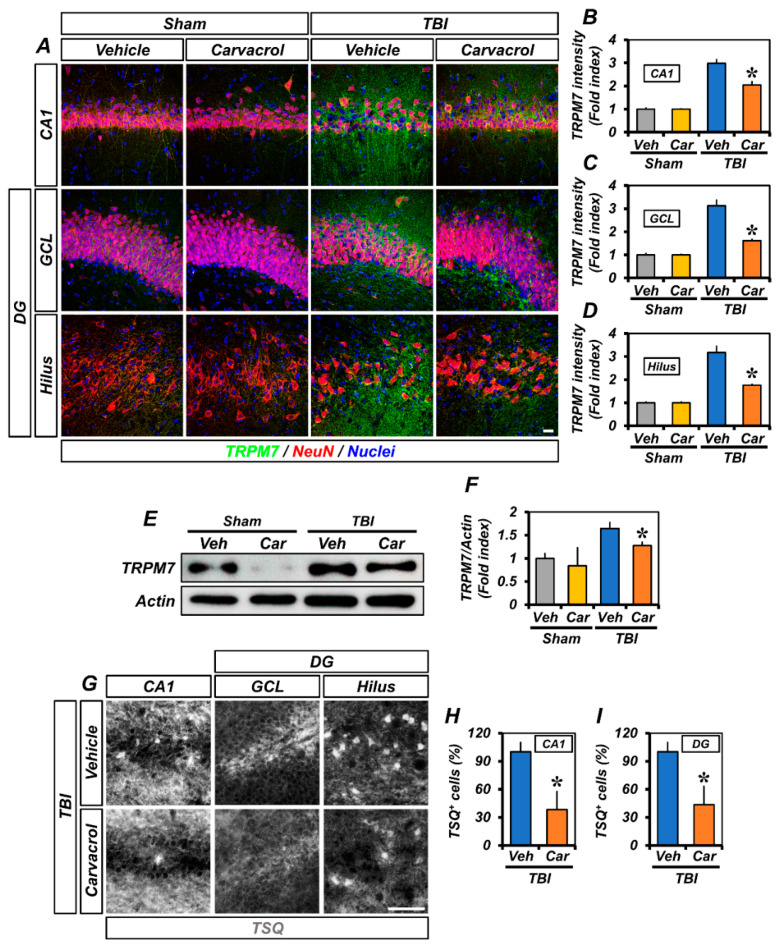
Carvacrol reduced TRPM7 over-expression and intracellular free zinc accumulation after TBI: (**A**) Double-label confocal micrographs representing neuronal marker NeuN^+^ neuronal nuclei (red) co-labeled with the TRPM7 (green) in the CA1, GCL, and hilus of ipsilateral hippocampus from vehicle- and carvacrol-treated groups 12 h after sham surgery or TBI. Nuclei are stained with DAPI (blue). Scale bar, 20 µm; (**B**–**D**) quantification of the immunofluorescence intensity of TRPM7 (green), as determined in the same CA1 (**B**), GCL (**C**), and hilus (**D**) regions of the ipsilateral hippocampus (mean ± SEM; *n* = 3–6 per group). * *p* < 0.05 vs. vehicle-treated TBI group (Kruskal–Wallis test followed by Bonferroni posthoc test: CA1: Chi square = 14.294, df = 3, *p* = 0.003; GCL: Chi square = 14.294, df = 3, *p* = 0.003; hilus: Chi square = 14.399, df = 3, *p* = 0.002); (**E**) Western blot of TRPM7 in the ipsilateral hippocampus; (**F**) bar graphs showing the quantification of TRPM7 protein amounts from the ipsilateral hippocampus (mean ± SEM; *n* = 3–4 per group). * *p* < 0.05 vs. vehicle-treated TBI group (Kruskal–Wallis test followed by Bonferroni post-hoc test: CA1: Chi square = 8.71, df = 3, *p* = 0.0334); (**G**) representative images showing sections of the ipsilateral hippocampus stained with TSQ for the detection of the accumulation of intracellular free zinc. Scale bar, 25 µm; and (**H**,**I**) bar graphs showing the number of TSQ^+^ neurons in the CA1 (**H**) and DG (**I**) of the ipsilateral hippocampus (mean ± SEM; *n* = 3 per group). * *p* < 0.05 vs. vehicle-treated TBI group (unpaired Student’s *t*-test).

**Figure 2 ijms-23-13840-f002:**
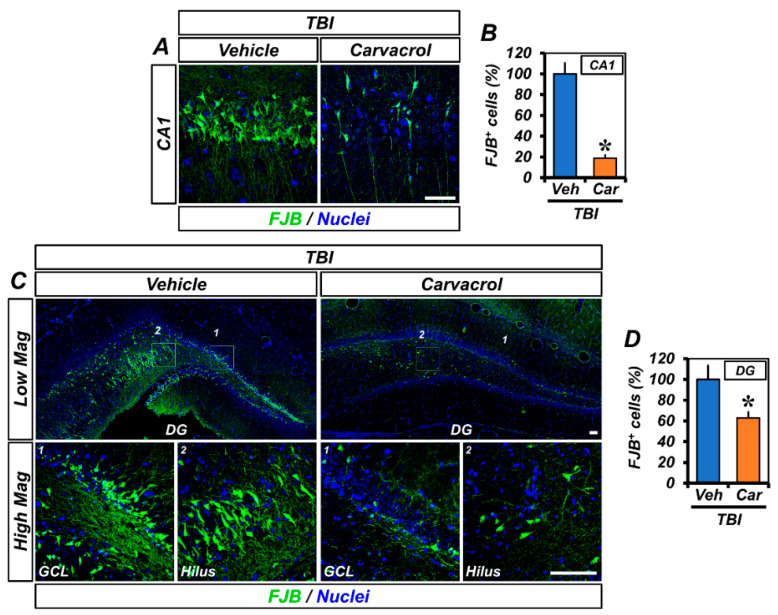
Carvacrol reduced neuronal degeneration after TBI: (**A**,**C**) Representative images showing sections of the hippocampal CA1 (**A**) and GCL and hilus of DG (**C**) stained with FJB to detect degenerating neurons. Nuclei are stained with DAPI (blue). Scale bar, 50 µm; and (**B**,**D**) quantification of the number of FJB^+^ cells from the CA1 (**B**) and DG (**D**) of the ipsilateral hippocampus (mean ± SEM; *n* = 5–6 per group). * *p* < 0.05 vs. vehicle-treated TBI group (unpaired Student’s *t*-test).

**Figure 3 ijms-23-13840-f003:**
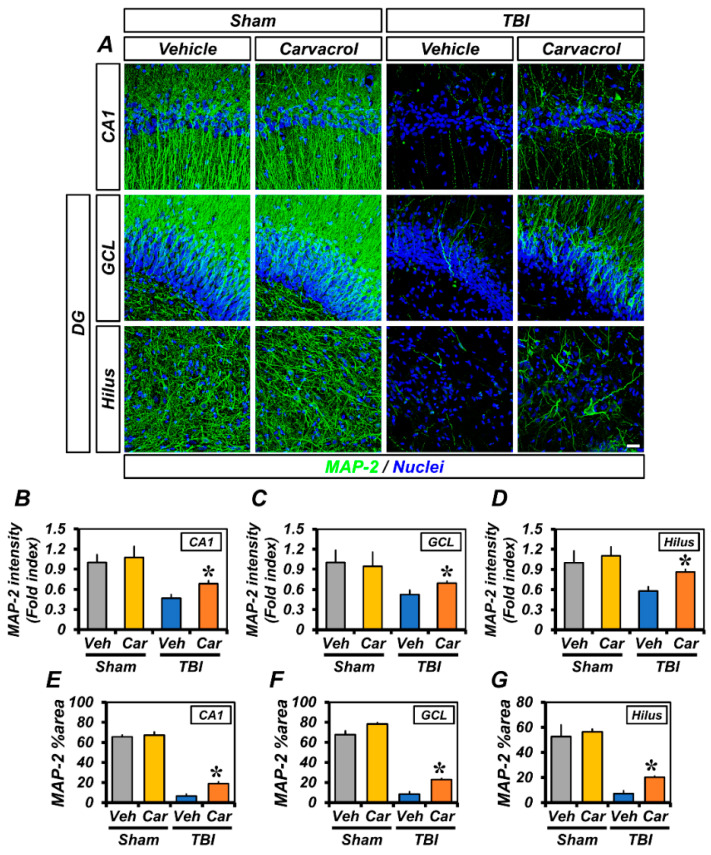
Carvacrol suppressed dendritic injury after TBI: (**A**) Representative immunofluorescence images showing sections of hippocampus stained with MAP-2 antibody to detect neuronal dendrites. Nuclei are stained with DAPI (blue). Scale bar, 20 µm; (**B**–**D**) quantification of the immunofluorescence intensity of MAP-2 (green) as determined in the same CA1 (**B**), GCL (**C**), and hilus (**D**) regions of the ipsilateral hippocampus with or without carvacrol treatment in sham-operated and TBI groups (mean ± SEM; *n* = 3–6 per group). * *p* < 0.05 vs. vehicle-treated TBI group (Kruskal–Wallis test followed by Bonferroni post-hoc test: CA1: Chi square = 13.437, df = 3, *p* = 0.004; GCL: Chi square = 8.354, df = 3, *p* = 0.039; hilus: Chi square = 10.315, df = 3, *p* = 0.016); and (**E**–**G**) bar graphs showing the percentage areas of MAP-2 immunoreactivity, as determined in the same CA1 (**E**), GCL (**F**), and hilus (**G**) regions of the ipsilateral hippocampus with or without carvacrol treatment in sham-operated and TBI groups (mean ± SEM; *n* = 3–6 per group). * *p* < 0.05 vs. vehicle-treated TBI group (Kruskal–Wallis test followed by Bonferroni post-hoc test: CA1: Chi square = 14.242, df = 3, *p* = 0.003; GCL: Chi square = 14.765, df = 3, *p* = 0.002; hilus: Chi square = 14.294, df = 3, *p* = 0.003).

**Figure 4 ijms-23-13840-f004:**
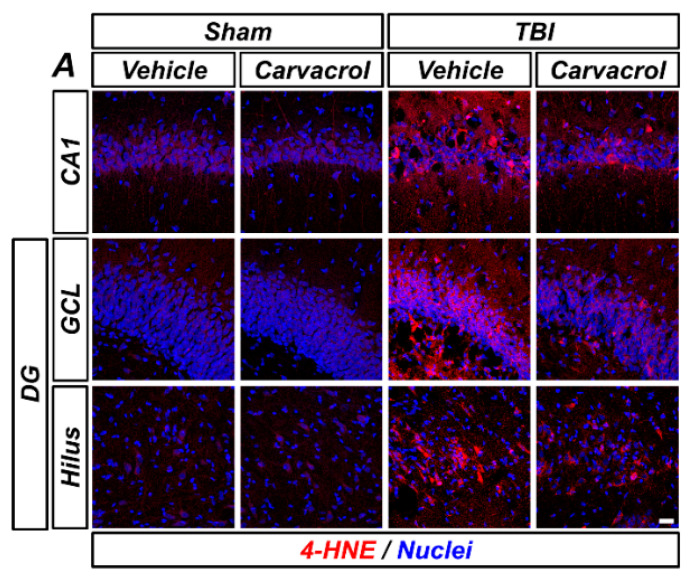
Carvacrol reduced oxidative stress and glutathione depletion after TBI: (**A**) Representative images of 4-HNE immunostaining in the hippocampal CA1 and GCL and the hilus of the DG to detect lipid peroxidation. Nuclei are stained with DAPI (blue). Scale bar, 20 μm; (**B**–**D**) quantification of 4-HNE immunofluorescence intensity in the CA1 (**B**), GCL (**C**), and hilus (**D**) regions of the ipsilateral hippocampus with or without carvacrol treatment in sham-operated and TBI groups (mean ± SEM; *n* = 3–6 per group). * *p* < 0.05 vs. vehicle-treated TBI group (Kruskal–Wallis test followed by Bonferroni post-hoc test: CA1: Chi square = 13.593, df = 3, *p* = 0.004; GCL: Chi square = 11.763, df = 3, *p* = 0.008; hilus: Chi square = 12.542, df = 3, *p* = 0.006); (**E**) representative images of nNOS (green) and nitrotyrosine (red) double immunostaining in the hippocampal CA1 to detect nitrosative stress. Nuclei are stained with DAPI (blue). Scale bar, 10 µm; (**F**,**G**) quantification of the immunofluorescence intensity of nNOS (**F**) and nitrotyrosine (**G**) immunoreactivity, as determined in the same CA1 region of the ipsilateral hippocampus (mean ± SEM; *n* = 3 per group). * *p* < 0.05 vs. vehicle-treated TBI group (Kruskal–Wallis test followed by Bonferroni post-hoc test: nNOS: Chi square = 9.462, df = 3, *p* = 0.024; nitrotyrosine: Chi square = 9.359, df = 3, *p* = 0.025); (**H**) double-labeled confocal micrographs of glutathione-N-ethylmaleimide (GS-NEM, green) and neuronal nuclei (NeuN, red) in the hippocampal CA1 to detect neuronal GSH. Nuclei are stained with DAPI (blue). Scale bar, 10 μm; and (**I**) bar graph showing the fluorescence intensity of GS-NEM immunoreactivity in the same CA1 region of the ipsilateral hippocampus (mean ± SEM; *n* = 3 per group). * *p* < 0.05 vs. vehicle-treated TBI group (Kruskal–Wallis test followed by Bonferroni post-hoc test: GS-NEM: Chi square = 9.359, df = 3, *p* = 0.025).

**Figure 5 ijms-23-13840-f005:**
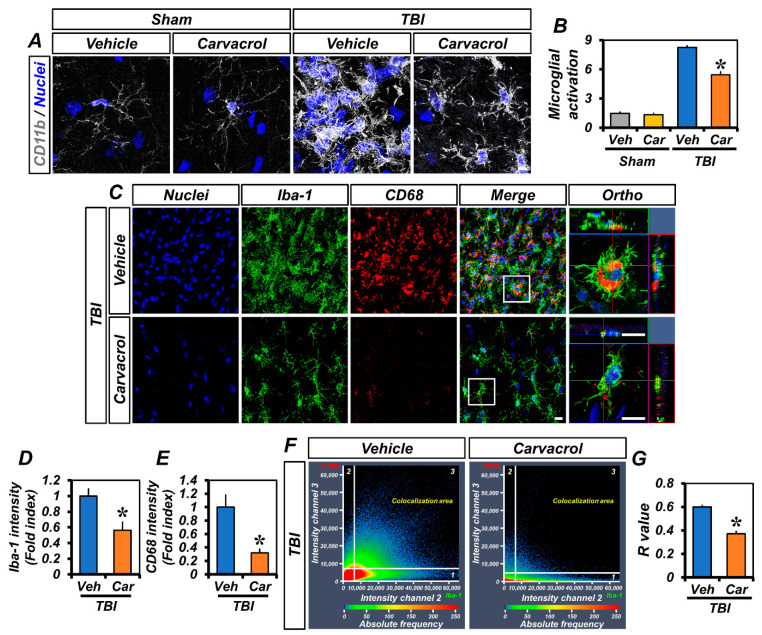
Carvacrol reduced microglial activation after TBI: (**A**) Immunofluorescence images representing the microglial activation in the hippocampus of sham-operated and TBI groups (either vehicle or carvacrol) at day 7 as shown by immunofluorescence staining for CD11b (grey). Nuclei are stained with DAPI (blue). Scale bar, 5 μm; (**B**) quantification of the grades of microglial activation as determined in the same hippocampal region (mean ± SEM; *n* = 3–6 per group). * *p* < 0.05 vs. vehicle-treated TBI group (Kruskal–Wallis test followed by Bonferroni post-hoc test: CA1: Chi square = 13.480, df = 3, *p* = 0.004); (**C**) representative images of Iba-1- (green) and CD68 (red)-immunopositive cells, as well as merged images, for vehicle- and carvacrol-treated TBI groups. Nuclei are stained with DAPI (blue). Scale bar, 50 µm; (**D**,**E**) bar graphs showing the immunofluorescence intensity of Iba-1 (**D**) and CD68 (**E**), as determined in the same hippocampal region (mean ± SEM; *n* = 3 per group). * *p* < 0.05 vs. vehicle-treated TBI group (unpaired Student’s *t*-test); (**F**) co-localization scatterplots for CD68 and Iba-1; and (**G**) Mander’s overlap coefficient (mean ± SEM; *n* = 3 per group). * *p* < 0.05 vs. vehicle-treated TBI group (unpaired Student’s *t*-test).

**Figure 6 ijms-23-13840-f006:**
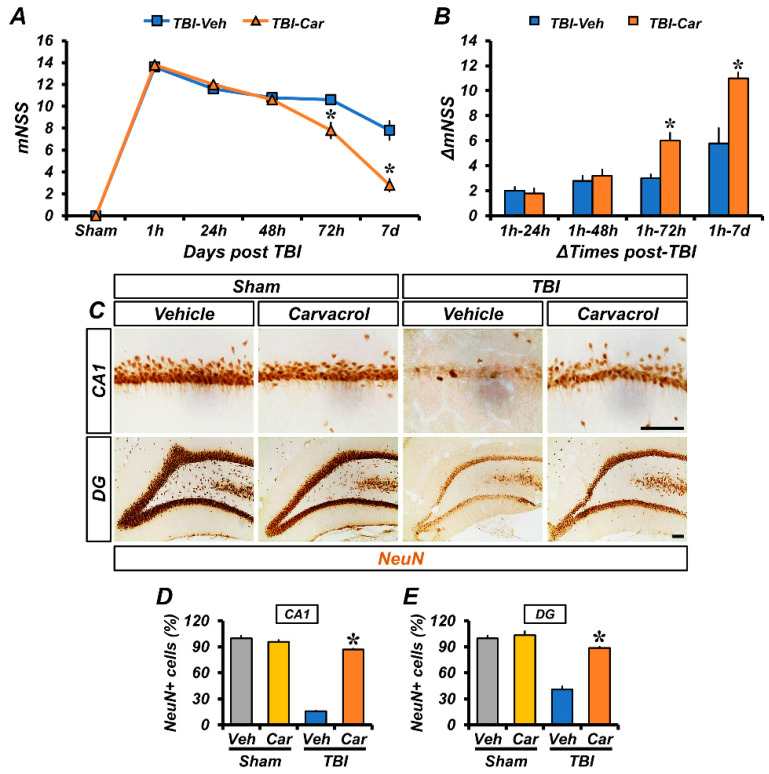
Carvacrol not only improved neurological function but also reduced neuronal death after TBI: (**A**) The mNSS was determined in rats at 1, 24, 48, 72 h, and 7 days after TBI. A score of 18 means that all tasks were failed; a score of 0 means that all tasks were successfully completed (mean ± SEM; *n* = 5 per group). * *p* < 0.05 vs. vehicle-treated TBI group (repeated measures ANOVA; Time: F = 583.733, *p* < 0.001; Group: F = 2.4, *p* = 0.016; Time × Group interaction: F = 8.882, *p* < 0.001); (**B**) changes in mNSS (ΔmNSS) were assessed at various time intervals between the 1 h and multiple pre-determined time points thereafter (mean ± SEM; *n* = 5 per group). * *p* < 0.05 vs. vehicle-treated TBI group (repeated measures ANOVA; Time: F = 74.780, *p* < 0.001; Group: F = 12.041, *p* = 0.008; Time × Group interaction: F = 14.363, *p* < 0.001); (**C**) photomicrographs showing sections of the hippocampal CA1 and DG stained for the neuronal marker NeuN, scale bar, 100 μm; (**D**,**E**) bar graphs showing the number of NeuN^+^ neurons in the hippocampal CA1 (**D**) and DG (**E**) areas (mean ± SEM; *n* = 3–6 per group). * *p* < 0.05 vs. vehicle-treated TBI group (Kruskal–Wallis test followed by Bonferroni post-hoc test: CA1: Chi square = 13.459, df = 3, *p* = 0.004; DG: Chi square = 13.243, df = 3, *p* = 0.004).

**Figure 7 ijms-23-13840-f007:**
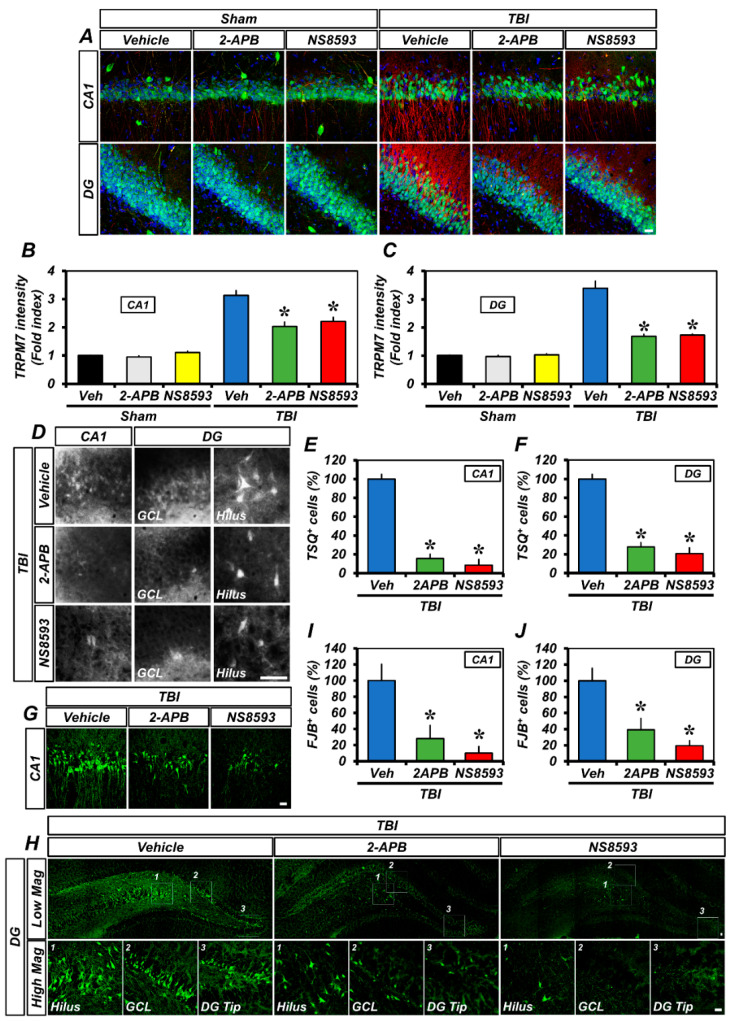
2-APB or NS8593 reduced TRPM7 over-expression, intracellular free zinc accumulation, and neuronal degeneration after TBI: (**A**) confocal micrographs representing the neuronal marker NeuN^+^ neuronal nuclei (green) co-labeled with the TRPM7 (red) in the CA1 and DG of the ipsilateral hippocampus from the vehicle-, 2-APB-, and NS8593-treated groups 12 h after sham surgery or TBI. Nuclei are stained with DAPI (blue). Scale bar, 20 µm; (**B**,**C**) quantification of the immunofluorescence intensity of TRPM7 (red) as determined in the same CA1 (**B**) and DG (**C**) regions of the ipsilateral hippocampus (mean ± SEM; *n* = 3–5 per group). * *p* < 0.05 vs. vehicle-treated TBI group (one-way analysis of variance (ANOVA) followed by Bonferroni post-hoc test; *F* = 39.76, *p* < 0.0001); (**D**) representative images showing sections of the ipsilateral hippocampus stained with TSQ for the detection of the accumulation of intracellular free zinc. Scale bar, 25 µm; (**E**,**F**) bar graphs showing the number of TSQ^+^ neurons in the CA1 (**E**) and DG (**F**) of the ipsilateral hippocampus (mean ± SEM; *n* = 4 per group). * *p* < 0.05 vs. vehicle-treated TBI group (one-way ANOVA followed by Bonferroni post-hoc test; (**E**): *F* = 93.91, *p* < 0.0001; (**F**): *F* = 20.46, *p* = 0.0004); (**G**,**H**) representative images showing sections of the hippocampal CA1 (**G**) and DG (**H**) stained with FJB to detect degenerating neurons. Scale bar, 20 µm; and (**I**,**J**) quantification of the number of FJB^+^ cells in the CA1 (**I**) and DG (**J**) of the ipsilateral hippocampus (mean ± SEM; *n* = 4–5 per group). * *p* < 0.05 vs. vehicle-treated TBI group (one-way ANOVA followed by Bonferroni post-hoc test; I: *F* = 9.459, *p* = 0.0041; J: *F* = 11.99, *p* = 0.0017).

**Figure 8 ijms-23-13840-f008:**
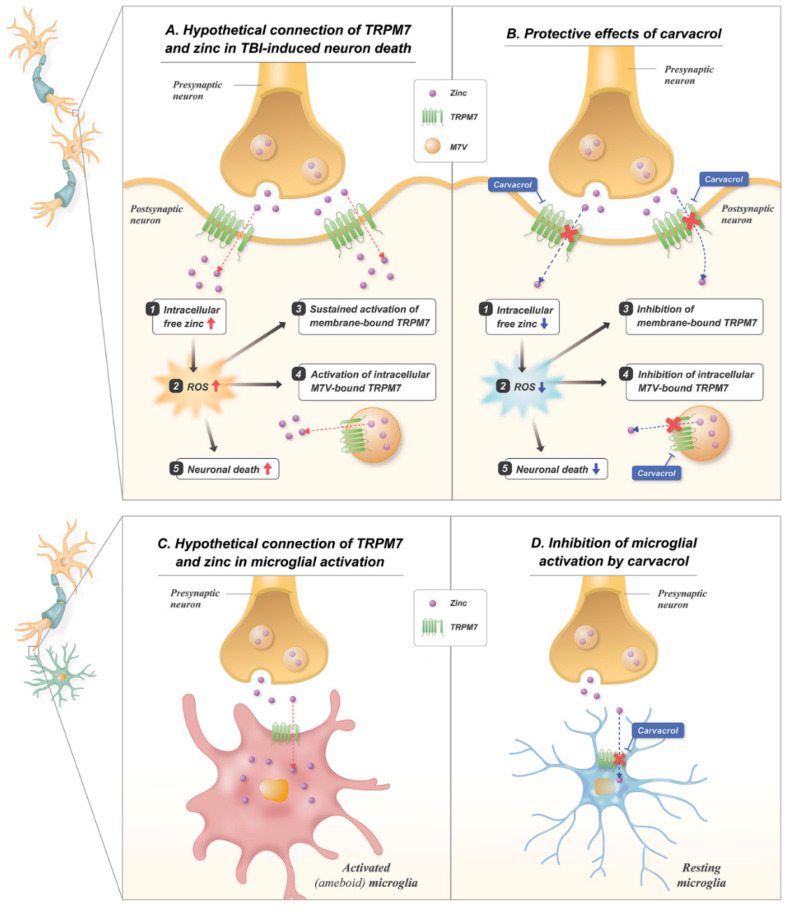
Proposed mechanisms for the effects of carvacrol on TBI-induced neuronal death and microglial activation. Schematic illustrations indicate the proposed action of carvacrol through inhibition of membrane-bound TRPM7 and intracellular vesicle-bound TRPM7 on neuronal death and microglial activation induced by TBI: (**A**) TBI induces the release of pre-synaptic zinc into the synapse, which translocates into the post-synaptic neurons through the TRPM7 channel. Increased zinc translocation contributes to the production of ROS. Increased ROS induces the activation of both membrane- and M7Vs-bound TRPM7, leading to the accumulation of free zinc in the cytosol, contributing to further ROS production. Eventually, zinc and ROS induce further neuronal injury; (**B**) carvacrol blocks the TRPM7 channels, leading to less translocation of zinc into the post-synaptic neurons after TBI. Inhibition of ROS production further leads to less activation of TRPM7, which is also blocked by carvacrol. Reduction in zinc reflux and ROS production reduces neuronal death; (**C**) synaptically released zinc gives rise to its influx into nearby microglia through TRPM7 channels, and this triggers microglial activation; (**D**) inhibition of TRPM7 by carvacrol can result in the suppression of zinc influx, thereby blocking microglial activation.

**Table 1 ijms-23-13840-t001:** Detailed serial scores of modified NSS in TBI-vehicle and TBI-carvacrol groups.

Time Point	Modified NSS	TBI-Vehicle	TBI-Carvacrol	*p*-Value
1 h	Motor tests	4.0 ± 0.45	4.2 ± 0.37	
	Sensory tests	1.2 ± 0.2	1.6 ± 0.24	
	Beam balance tests	6.0 ± 0.0	6.0 ± 0.0	
	Reflexes absent and abnormal movements	2.4 ± 0.24	2.0 ± 0.0	
	Total points	13.6 ± 0.4	13.8 ± 0.2	0.671
24 h	Motor tests	4.0 ± 0.2	3.6 ± 0.24	
	Sensory tests	1.0 ± 0.0	1.2 ± 0.2	
	Beam balance tests	5.8 ± 0.2	6.0 ± 0.0	
	Reflexes absent and abnormal movements	1.0 ± 0.32	1.2 ± 0.2	
	Total points	11.6 ± 0.24	12 ± 0.32	0.348
48 h	Motor tests	3.4 ± 0.24	3.0 ± 0.0	
	Sensory tests	1.2 ± 0.2	1.0 ± 0.0	
	Beam balance tests	5.4 ± 0.24	5.4 ± 0.4	
	Reflexes absent and abnormal movements	0.8 ± 0.2	1.2 ± 0.2	
	Total points	10.8 ± 0.2	10.6 ± 0.4	0.671
72 h	Motor tests	3.0 ± 0.32	2.2 ± 0.37	
	Sensory tests	1.0 ± 0.32	0.6 ± 2.4	
	Beam balance tests	5.6 ± 0.24	3.8 ± 0.49	
	Reflexes absent and abnormal movements	1.0 ± 0.0	1.2 ± 0.2	
	Total points	10.6 ± 0.4	7.8 ± 0.73	0.015
7 days	Motor tests	2.2 ± 0.37	1.0 ± 0.0	
	Sensory tests	0.8 ± 0.2	0.0 ± 0.0	
	Beam balance tests	3.8 ± 0.58	1.2 ± 0.37	
	Reflexes absent and abnormal movements	1.0 ± 0.0	0.6 ± 0.24	
	Total points	7.8 ± 0.86	2.8 ± 0.58	0.002

Data are expressed as the mean ± standard deviation. Higher scores represent a more severe neurological deficit. The mNSS scores were compared using the Wilcoxon rank-sum test. Abbreviations: NSS, Neurological Severity Scale.

**Table 2 ijms-23-13840-t002:** Effects of TRPM7 inhibition in multiple acute neurological injuries.

Disease Models	Species	TRPM7 Inhibitors	Divalent Ions	Immunohistochemistry	Neurological Assessment	Reference
TBI	Rats	Carvacrol, 2-APB	Zn^2+^↓	TRPM7 expression↓ Degenerating neurons↓ Dendritic injury↓ Oxidative/Nitrosative stress↓ Glutathione depletion↓ M1 microglial activation↓ Neuronal death↓	Neurological dysfunction↓	
Stroke	Rats	Carvacrol,	Zn^2+^↓	TRPM7 expression↓ Degenerating neurons↓ Dendritic injury↓ Oxidative stress↓ Microglial activation ↓	N/A	[35]
Epilepsy	Rats	Carvacrol, 2-APB	Zn^2+^↓	TRPM7 expression↓ Degenerating neurons↓ Dendritic injury↓ Oxidative stress↓ M1 microglial activation↓ BBB disruption↓ Apoptosis↓ Neuronal death↓	N/A	[36]

Abbreviations: TBI, Traumatic Brain Injury; TRPM7, transient receptor potential melastatin 7; 2-APB, 2-Aminoethoxydiphenyl borate; BBB, Blood–Brain Barrier.

## Data Availability

The data supporting the presented findings are freely available from the corresponding author upon reasonable request.

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
