# Peer review of "Carvacrol Inhibits Expression of Transient Receptor Potential Melastatin 7 Channels and Alleviates Zinc Neurotoxicity Induced by Traumatic Brain Injury"

_ijms, 2022, doi:10.3390/ijms232213840_

Round 1
Reviewer 1 Report (Previous Reviewer 2)
The article “Carvacrol Inhibits Expression of Transient Receptor Potential 2 Melastatin 7 Channels and Alleviates Zinc Neurotoxicity Induced by Traumatic Brain Injury” by Lee and coworkers describes the effects of carvacrol on traumatic brain injury-induced zinc neurotoxicity and the effects on TRPM7 expression. The study is well designed. The experiments are conclusive and scientifically sound. The conclusions are supported by the results and critically discussed based on the current literature. The ms. is well written and a pleasure to read. Therefore, I recommend the ms for publication after a few minor revisions.
Minor revisions:
Line 60-61: The sentence needs to be corrected according to citation [14] like this: A previous study has shown that carvacrol inhibited the function of TRPM7 ectopically expressed in a primary culture of hippocampal cornu ammonis (CA)1-3 neurons.
Line 387-388: In this context the sentence makes no sense! I guess the authors would like to say: “To confirm and further strengthen our hypothesis, the development of a specific TRPM7 inhibitor is needed.
Author Response
Reviewer #1: The article “Carvacrol Inhibits Expression of Transient Receptor Potential Melastatin 7 Channels and Alleviates Zinc Neurotoxicity Induced by Traumatic Brain Injury” by Lee and coworkers describes the effects of carvacrol on traumatic brain injury-induced zinc neurotoxicity and the effects on TRPM7 expression. The study is well designed. The experiments are conclusive and scientifically sound. The conclusions are supported by the results and critically discussed based on the current literature. The ms. is well written and a pleasure to read. Therefore, I recommend the ms for publication after a few minor revisions.
Minor comments
Line 60-61: The sentence needs to be corrected according to citation [14] like this: A previous study has shown that carvacrol inhibited the function of TRPM7 ectopically expressed in a primary culture of hippocampal cornu ammonis (CA)1-3 neurons.
<Response: We appreciate this reviewer’s comments and agree with this reviewers point. As this reviewer suggested, we corrected it in the revised manuscript.>
Line 387-388: In this context the sentence makes no sense! I guess the authors would like to say: “To confirm and further strengthen our hypothesis, the development of a specific TRPM7 inhibitor is needed.
<Response: We appreciate this reviewer’s comments and agree with this reviewers point. As this reviewer suggested, we corrected it in the revised manuscript.>

Reviewer 2 Report (New Reviewer)
The Idea and data presented by author seems seem interesting and manuscript written well.. The experiments performed well and the data presented is clear to understand. The manuscript is acceptable for publication but need some minor revision particularly the following important points need to address before publication.
1) Manuscript language can be improved.
2) Figure 6C need to be label properly as Sham and TBI missing there.
3) In the discussion part, on page 12, line number: 342 to 344 need citation.
4) Check typing, spacing etc. mistakes
Some questions:
1. Why the apoptotic markers were not been checked?
2. Why other brain Areas not included in the study?
Suggestion:
Could be interesting to include Cresol Violate staining results

Author Response
Reviewer #2: The Idea and data presented by author seems seem interesting and manuscript written well.. The experiments performed well and the data presented is clear to understand. The manuscript is acceptable for publication but need some minor revision particularly the following important points need to address before publication.
1) Manuscript language can be improved.
<Response: Thank you for your suggestion. This revised manuscript is edited by MDPI English Editing company (ID: English-51194). We attached editing certificate at the end of the response.>
2) Figure 6C need to be label properly as Sham and TBI missing there.
<Response: We appreciate this reviewer’s comments and apologizes there was a mistake. Thus, we corrected it in the revised manuscript and added this at the end of the response.>
3) In the discussion part, on page 12, line number: 342 to 344 need citation.
<Response: We appreciate this comment. As this reviewer’s suggestion, we added the relevant reference in the revised manuscript.
4) Check typing, spacing etc. mistakes
<Response: Thank you for your suggestion. Even though our manuscript was edited by MDPI English Editing company, we once again checked typing, spacing and any mistakes and corrected it in the revised manuscript.
Some questions:
- Why the apoptotic markers were not been checked?
<Response: Thank you for your insightful comments. As the reviewer has asked, various cascades of neuronal death including apoptosis occurs after controlled cortical impact models. However, our research question was not focused on in which cascade the carvacrol may have an impact on neuronal death after TBI. Rather, we aimed to evaluate whether the administration of carvacrol, the TRPM7 inhibitor, can alleviate the whole process of neuronal death via blockage of excessive zinc translocation. Thus, we performed FJB staining in the earlier period after TBI for the evaluation of degenerating neurons and NeuN immunohistochemistry for the detection of alive neurons in the subacute stage. However, we agree that your point is important and we may include apoptotic markers in the future studies with TBI models.
- Why other brain areas not included in the study?
<Response: The controlled cortical impact model we have utilized in this study specifically induces consistent damage to CA1, DG and other areas of hippocampus. As other brain areas including cortex are directly injured by cortical impact, it is inadequate to evaluate neuronal damage with high reproducibility.
Suggestion:
Could be interesting to include Cresol Violate staining results
<Response: We appreciate this reviewer’s comments and agree with this reviewer’s point. It will be a very interesting study to verify the difference between vehicle and carvacrol in the extent of the TBI-induced lesions using cresyl violet staining. It would be a very good suggestion. However, due to time limitation for this revision, we plan for it to be part of our future experimental plans following the present work.>

This manuscript is a resubmission of an earlier submission. The following is a list of the peer review reports and author responses from that submission.
Round 1
Reviewer 1 Report
General comments: In this manuscript, Lee et al. use carvacrol as an inhibitor of TRPM7 to counteract neurotoxicity induced by traumatic brain injury in rat. The manuscript is well organized and the data are convincing. My main criticisms are 1) immunofluorescence is only technique for studying TRPM7 expression and 2) the use of carvacrol and 2-APB as TRPM7 blockers. Therefore, I believe that additional experiments are needed to improve the quality of this work.
Major comments:
- The results on TRPM7 expression following carvacrol treatment should be confirmed by western blotting and by qRT-PCR.
- Carvacrol and 2-APB are not selective blockers of TRPM7 channels because they are well known to also inhibit other TRPs or Orai channels. The authors should use NS8593 instead of 2-APB to confirm the effect of carvacrol on TRPM7 expression and zinc influx.
- The numeric data may be presented as percentages normalized to sham in order to facilitate the reading.
- Sham conditions is missing in some results like TSQ and FJB staining as well as for the mNSS score calculation.
- How is the MAP2 %area determined?
- TRPM7 expression and its inhibition by carvacrol during the microglial activation should be shown.
Minor comments:
- Spaces are sometime missing before references.
- Latin text should be in italics.
Reviewer 2 Report
The ms “Inhibition of Transient Receptor Potential Melastatin 7 by Carvacrol Alleviates Zinc Neurotoxicity Induced by Traumatic 3 Brain Injury “ by Lee et el. documents the third study from the same group of authors that describes the effects of carvacrol in neurological disorders with a similar outcome. The study is well designed, the experiments are conclusive and scientifically sound. The ms is well written and was a pleasure to read.
Main criticism
However, I need to express my doubts about the main conclusion of the manuscript which is not sufficiently supported by the data. The authors claim already in the title of the ms. that there is a causal relationship between the inhibition of TRPM7 ion channels by carvacrol and a number of secondary effects related to carvacrol treatment after TBI, like reduction of neuronal degeneration, oxidative stress, microglia activation, or the improvement of neurological impairment. To support this statement the authors first show by immunohistochemistry a coinciding downregulation of TBI-induced TRPM7 expression after carvacrol treatment (fig 1 A-D) and second that another TRPM7 inhibitor 2-APB has very similar effects upon the neuronal phenotype and TRPM7 expression (fig. 7 A-C). These findings indeed suggest somehow, that there is a link between TRPM7 proteins and the carvacrol effects on TBI, but they do not prove it! It has been shown by others, that both carvacrol and 2-APB inhibit TRPM7 channels directly, but these compounds act rather unselective and there are a number of other Zn2+ permeable channels and ion exchangers that are also affected. Furthermore, it is completely unknown whether and how the expression of TRPM7 proteins is linked to TRPM7 channel activity. Thus it is can e.g. not be ruled out that the observed carvacrol-dependent downregulation of TRPM7 expression is secondary to the inhibition of Zinc influx via other pathways and I need to ask: Is this downregulation limited to TRPM7?
In the introduction, the authors claim (lines 57-58) that “ a previous study showed that carvacrol blocked the overexpression of TRPM7 at the hippocampal cornu ammonis (CA) regions [14].” This is not correct! In the mentioned publication by Parnas et al (2009) the authors showed that “carvacrol inhibits recombinant TRPM7 channel overexpressed in HEK cells” and “examined the effect of carvacrol on the function of (recombinant) TRPM7 ectopically expressed in primary culture of hippocampal CA1–CA3 neurons”. They “demonstrated that carvacrol inhibits the function of mammalian TRPM7 expressed in synaptic terminals of hippocampal neurons” but they did not analyze any effect of carvacrol on TRPM7 expression!
Similar experiments using TRPM7-deficient animals would indeed be more than helpful to confirm the authors´ hypothesis. However, it has already been shown that deletion of TRPM7 in whole animals is not compatible with life. I therefore strongly suggest that the authors adapt the title of their manuscript (e.g “Carvacrol inhibits expression of Transient Receptor Potential Melastatin 7 channels and alleviates Zinc Neurotoxicity Induced by Traumatic 3 Brain Injury”) and qualify all their statements in the text concerning the causal relationship between the effects of carvacrol on TRPM7 and neurotoxicity. Furthermore, alternative Zn2+ entry pathways as the origin of TBI-induced neurotoxicity should be discussed.
Minor comments
- In the discussion (line 338-340) the authors state that “After TBI, excessive free zinc ions translocate into post-synaptic neurons via cation channels and multiple transporters [32]” However, in the mentioned reference (Levenson, 2020) I could not find such a statement. Instead, it is mentioned that “Upon excitation, presynaptic neurons release vesicular zinc into the synaptic cleft between neurons where it binds and modulates a variety of postsynaptic receptors and channels such as the glutamate receptors N-methyl-d-aspartate(NMDA), -amino-3-hydroxy-5-methyl-4-isoxazolepropionic acid (AMPA), and kainate receptors, as well as glycine, -aminobutyric acid-A (GABAA) receptors, and voltage-gated calcium channels”(Levenson, 2020). The authors should revise this statement or cite (Leng, Shi, Xiong & Sun, 2014).
Leng T, Shi Y, Xiong ZG, & Sun D (2014). Proton-sensitive cation channels and ion exchangers in ischemic brain injury: new therapeutic targets for stroke? Prog Neurobiol 115: 189-209.
Levenson CW (2020). Zinc and Traumatic Brain Injury: From Chelation to Supplementation. Med Sci (Basel) 8.
- The links to the figures should be introduced earlier in the text. The link to table 2 is missing completely.